# Factorial structure, validity, and gender invariance of the UCLA-R loneliness scale in ecuadorian adolescents

**Wilson Guillermo Siguenza Campoverde** *, **Katy Maricela Chamba Leiva**

Department of Psychology, Universidad Técnica Particular de Loja, Loja, Ecuador

* wgsiguenza@utpl.edu.ec

## Abstract

### Background

Loneliness has a significant impact on mental and physical health across different stages of development, with particularly evident effects during adolescence. During this period, young individuals undergo substantial social and emotional transformations, making loneliness a global concern.

### Objective

This study aimed to analyze the psychometric properties of the UCLA-R Loneliness Scale in Ecuadorian adolescents through exploratory and confirmatory factor analysis, assess its internal consistency, and examine factorial invariance across genders.

### Method

The sample consisted of 718 school-aged adolescents (288 males and 430 females) aged 14–17 years (M = 15.72, SD = 0.747), selected through probabilistic sampling. A sociodemographic ad hoc questionnaire and the Spanish version of the 20-item UCLA-R Loneliness Scale were administered. The sample was randomly divided into two equivalent subsamples (n = 359) to separately perform an Exploratory Factor Analysis (EFA) and a Confirmatory Factor Analysis (CFA). The EFA was conducted using principal axis factoring, oblique rotation, and polychoric correlations. The CFA evaluated one-factor, two-factor, and three-factor models, employing fit indices such as CFI, TLI, RMSEA, and SRMR. Internal consistency was estimated using Cronbach's alpha and McDonald's omega coefficients. Additionally, factorial invariance analyses by gender were performed, along with a univariate ANOVA to examine potential gender differences.

in any medium, provided the original author and source are credited.

**Data availability statement:** All data from this study are publicly available under a CC0 1.0 Universal License in the Open Science Framework (OSF) repository. The dataset, titled 'UCLA-R Loneliness Scale Refined Database for Adolescents,' can be accessed at: https://osf.io/9zw43/files/osfstorage/682fc4f54cf1506f4b-b06c34 This includes: • The refined UCLA-R Loneliness Scale responses • Demographic data for adolescent participants The permanent DOI for this dataset is: DOI 10.17605/OSF.IO/9ZW43.

**Funding:** The author(s) received no specific funding for this work.

**Competing interests:** NO authors have competing interests.

## Results

The exploratory factor analysis (EFA) results indicated the presence of two factors, whose structure explains 40.8% of the total variance, with 19 items. The confirmatory factor analysis (CFA) confirmed that the bifactor model with 19 items exhibited a superior fit compared to the unidimensional and three-factor models with 20 items, with fit indices falling within acceptable to excellent ranges (CFI = .936, TLI = .927, RMSEA = .050, SRMR = .048). Additionally, the bifactor model demonstrated measurement invariance across genders. Regarding internal consistency, the scale demonstrated a Cronbach's alpha of .876 and an omega coefficient of .83, confirming its reliability.

## Conclusion

It is concluded that the 19-item UCLA-R Loneliness Scale is a valid and reliable instrument for assessing perceived loneliness in Ecuadorian adolescents, regardless of gender. Future research could replicate these findings in other regions and cultural contexts to validate its use on a larger scale.

## Introduction

Loneliness, as a psychological construct, influences the perception and evaluation of social networks, as well as the support received, leading to psychological distress and emotional emptiness [1]. From both an objective and subjective perspective, living alone or feeling lonely has been identified as a risk factor for physical and mental health [2–4]. Additionally, loneliness is considered a public health threat [5], affecting various age groups, including children, adolescents, and older adults [6–10]. Individuals experiencing loneliness face an increased risk of compromised physical and mental well-being [11–13]. It has been associated with childhood social isolation, problematic alcohol use, depression, and general health issues during adolescence [14,15]. Its impact is so profound that it extends across different geographical contexts [16–19].

In line with the above, Antunes et al. [20], Doan et al. [21], Garnow et al. [22], Marquez et al. [23], and Cosenza et al. [24] support the notion that adolescents are the most affected by this phenomenon. As a transitional stage characterized by peer interactions [25,26], adolescence is particularly vulnerable. In this regard, Lamash et al. [27] emphasize that social bonding serves as a protective factor for mental health, resilience, and well-being. However, the inability to establish positive social connections negatively impacts affect, life satisfaction [28], and feelings of emptiness [1].

In this way, loneliness is understood as an unpleasant subjective experience due to the discrepancy between an individual's desired and actual social needs [29], which can lead to stroke, anxiety, dementia, depression, suicide, among other conditions [5], carrying significant implications for morbidity and mortality rates [4]. In this context, having updated and validated instruments to accurately assess this construct becomes crucial [30–32].

It is thus that various scales have emerged to quantify the perception of loneliness. Among them are: the original version of the UCLA Loneliness Scale [33] and its revised version [34], the De Jong Gierveld and Kamphuis Loneliness Scale [35], the Loneliness and Aloneness Scale for Children and Adolescents (LACA) [36], the Social and Emotional Loneliness Scale for Adults (SESLA) [37] and its short version, the SELSA-S [38], the ESTE Loneliness Measurement Scale [39], the Perth-A Loneliness Scale [40], among others. Most of these scales are designed to assess loneliness in adults, and very few focus on evaluating loneliness in adolescents, even fewer in Hispanic adolescents [30].

In concordance with the aforementioned, several authors agree that the UCLA Loneliness Scale is the most widely used to assess loneliness [17,41–43]. Its original 20-item version has demonstrated adequate reliability measures (alpha coefficient = 0.96) and a test-retest correlation over a two-month period of 0.73 [33]. This scale enjoys broad recognition and has been adapted and validated in numerous countries and cultures, which demonstrates its versatility in various contexts. For this reason, short versions of the UCLA Loneliness Scale have been developed, some consisting of 11 items [44], ten items [45], eight items [46], six items [42], and three items [47]. Most of them have shown high levels of reliability [17,43]; however, these have been validated and used to assess loneliness in older adults [48] through telephone-based methods [47].

In line with the above, the abbreviated versions have demonstrated varied factorial structures. For example, the 11-item version revealed a two-factor model, assessing social connections and a sense of belonging in older adults [44]. The 8-item version, examined in Dutch adolescents, presented a three-factor model comprising loneliness, positive elements, and negative elements [49]. The 6-item version demonstrated a unidimensional structure for the Peruvian adolescent population [30]. Meanwhile, the original 20-item version includes a general loneliness factor and two uncorrelated measurement factors (negative and positive items) [45]. Regarding the 6-item version developed by Neto in 1992, it was found to be unidimensional, with a Cronbach's alpha coefficient of 0.77. Finally, the three-item version, considered an ultra-short form, showed lower Cronbach's alpha coefficients compared to longer scales [47].

In this regard, Fonsêca et al. [50] studied the validity and accuracy of the UCLA scale in 234 university students, demonstrating high reliability with a Cronbach's alpha of 0.93. Ausín et al. [41] validated the UCLA scale in an adult population aged between 65 and 84 years, using a sample of 409 participants. A confirmatory factor analysis was conducted to assess the factorial structure of the UCLA LS-R, and the scale's internal consistency yielded a Cronbach's alpha of 0.85. Carreño-Moreno et al. [51] validated the scale in caregivers of patients with chronic diseases in the Colombian population. The instrument underwent psychometric validation, including face and content validation, concluding that the scale has acceptable face and content validity for use in caregivers of individuals with chronic illness in Colombia. Alsubheen et al. [52] validated the UCLA scale in patients with chronic obstructive pulmonary disease, with findings demonstrating convergent and divergent validity, as well as test-retest reliability, in a population aged between 65 and 84 years.

In Ecuador, the UCLA-R scale has been widely used, particularly among university students, but its application in adolescents remains limited. However, to date, no formal validation process of this instrument has been conducted in the Ecuadorian context with adolescents. In this sense, the cross-cultural validation of this scale is essential to ensure that the constructs are measured accurately and equivalently across cultural contexts. For Ecuadorian adolescents, the use of a locally validated version of the UCLA-R scale ensures greater relevance, precision, and cultural sensitivity in the interpretation of results. Therefore, the absence of prior validations of the instrument in adolescents within the Ecuadorian context underscores the need to validate the UCLA-R Loneliness Scale through exploratory and confirmatory factor analysis, assess its internal consistency, and examine factorial invariance across genders, in Ecuadorian adolescents, ensuring an instrument adjusted to the country's sociocultural characteristics.

## Materials and methods

### Participants

The study sample was selected using simple random sampling, as described by Cohen and Manion [53]. This approach ensured that each adolescent in the defined population had an equal probability of being selected, which enhanced the

representativeness of the sample and increased the potential for generalizing the findings. This selection was based on data provided by the VIII Population Census and VII Housing Census of 2022 in the Republic of Ecuador, conducted by the National Institute of Statistics and Censuses [54], which reported a total population of 1,903,170 adolescents within the age range of 12–17 years.

The final random sample consisted of 718 school-attending adolescents, with a distribution of 288 males and 430 females. Participants' ages ranged from 14 to 17 years (M = 15.72 years, SD = 0.747 years). The validity of the sample was supported by a 95% reliability level, with a 5% margin of error in its representation of the total population, as established by Buendía Eisman [55]. The demographic data of the participants are presented in Table 1.

## Instruments

**Ad hoc sociodemographic questionnaire.** Designed to collect data on age, sex, current educational level, cohabitation status, father's and mother's age, and self-perceived socioeconomic level.

**UCLA-R loneliness scale.** The Spanish translation by Vera et al. [30] was used. The scale consists of 20 items designed to assess loneliness in various populations. Of these items, 11 are negatively worded (2, 3, 4, 7, 8, 11, 12, 13, 14, 17, 18), and nine are positively worded (1, 5, 6, 9, 10, 15, 16, 19, and 20). The scale is answered using a Likert-type format with response options: 4 = frequently, 3 = sometimes, 2 = rarely, and 1 = never. The total loneliness score is obtained by summing all 20 items. The score range extends from 20 to 80, with higher scores indicating a greater level of loneliness. The complete scale demonstrated satisfactory internal consistency (α = .96); however, for this study, an α = .876 and an omega coefficient (ω) of .83 were obtained.

**Table 1. Demographic characteristics of the participants.**

| Variable | Mean | SD | Min/Max | n | % |
|---|---|---|---|---|---|
| Age | 15.727 | .747 | 14/ 17 | | |
| **Sex** | | | | | |
| Male | | | | 288 | 40.11 |
| Female | | | | 430 | 59.89 |
| **Educational Level** | | | | | |
| First year of high school | | | | 374 | 52.09 |
| Second year of high school | | | | 344 | 48.00 |
| **With Whom They Live** | | | | | |
| Father | | | | 16 | 2.23 |
| Mother | | | | 80 | 11.14 |
| Siblings | | | | 6 | 0.84 |
| Father and Siblings | | | | 24 | 3.34 |
| Mother and Siblings | | | | 161 | 22.42 |
| Father, Mother, and Siblings | | | | 379 | 52.79 |
| Others | | | | 52 | 7.24 |
| **Father's Age** | 44.953 | 7.82 | 28/ 65 | | |
| **Mother's Age** | 41.669 | 7 | 29/ 65 | | |
| **Socioeconomic Level** | | | | | |
| High | | | | 33 | 4.60 |
| Middle | | | | 634 | 88.30 |
| Low | | | | 51 | 7.10 |

Note: SD, standard deviation; Min, minimum value; Max, maximum value; n, absolute frequency of the sample; %, percentage.

## Procedure

For this study, coordination was made with the authorities and parents of four educational institutions in Loja, Ecuador. Parents were informed about the procedure and objectives of the research, and their authorization was requested for their children's participation (physical informed consent). Similarly, a detailed explanation of the study was provided to the adolescents, who participated freely and voluntarily (physical informed assent).

Regarding data collection, the UCLA-R Loneliness Scale and the Ad hoc sociodemographic survey were digitized. Google Forms, an online survey administration tool by Google, was used. The survey was available for participants from May 29 to June 29, 2023. To minimize potential respondent bias, particularly in online surveys, several strategies were implemented. First, data were collected in a controlled setting (the computer lab of each educational institution) under research team supervision, ensuring independent participation without external influence. Second, the study objectives and questionnaire procedures were explained in detail, both verbally and in writing, with assistance provided as needed. Third, written informed consent and assent were obtained prior to participation, emphasizing its voluntary nature. Although no formal online survey reporting guideline was followed, the procedure combined digital data collection with in-person supervision to ensure data consistency, clarity, and integrity

## Ethical considerations

To ensure minimal risk to participants and comply with ethical principles in research involving human subjects, the study received ethical approval from the Universidad Central del Ecuador, under Code 002-EXT-2023. The study adhered to the ethical standards of the 1964 Declaration of Helsinki and its subsequent amendments [56].

## Data analysis

For statistical analyses, the IBM Statistical Package for the Social Sciences (SPSS) (IBM Inc., Chicago, IL, USA; version 26.0) and Jeffreys's Amazing Statistics Program (JASP, version 0.18.3; JASP Team, University of Amsterdam) were used.

The factorial structure of the scale was identified through exploratory factor analysis (EFA) and confirmatory factor analysis (CFA). Following Harrington [57], both EFA and CFA should be conducted on separate samples. The total sample (N = 718) was randomly split into two independent and equal subsamples (nA = 359) and (nB = 359). The chi-square test did not reveal significant differences between the subsamples, indicating that random selection maintained a balanced proportion of sociodemographic characteristics in both groups.

In the EFA, the suitability of the data matrix was assessed using the Kaiser-Meyer-Olkin (KMO) measure of sampling adequacy. Additionally, Bartlett's test of sphericity was conducted. To assess data normality beyond univariate skewness and kurtosis, Mardia's kurtosis coefficient was employed. According to Mardia et al. [58], if the critical value is lower than 1.96, the data can be considered normally distributed at a significance level of 0.05. Based on this criterion, the parallel analysis retention method, using principal components, was applied [59].

Regarding the extraction method, principal axis factoring with Oblimin oblique rotation was applied, following the recommendations of Lloret-Segura et al. [60]. This approach had been previously used by Lasgaard [61] in the reliability and validity analysis of the Danish version of the UCLA Loneliness Scale with adolescents. Additionally, the polychoric/tetrachoric correlation matrix was utilized to more accurately capture relationships between categorical and ordinal variables. Factor loadings of ≥.30 were considered acceptable, in line with Hinkin [62,63].

Subsequently, a confirmatory factor analysis (CFA) was performed, employing the following fit indices: Bentler's Comparative Fit Index (CFI), Tucker-Lewis Index (TLI), Root Mean Square Error of Approximation (RMSEA), and Standardized Root Mean Square Residual (SRMR). Additionally, the chi-square ratio ($\chi^2$/df) or minimum discrepancy per degree of freedom (CMIN/DF), Akaike Information Criterion (AIC), and Bayesian Information Criterion (BIC) were included.

For model fit evaluation, the following thresholds were applied: CMIN/DF ≤ 3 is acceptable, ≥ 2 is optimal [64]; CFI and TLI ≥ .90 are acceptable, ≥ .95 are optimal; RMSEA and SRMR ≤ .08 are acceptable, ≤ .05 are optimal [65]. AIC and BIC

were used to compare alternative models, where the lowest value in each case indicates the best model fit [64]. The internal consistency of the scale was assessed using the omega coefficient (ω) and Cronbach's alpha (α).

Regarding factorial invariance, the configural invariance model (MC), metric invariance model (MM), scalar invariance model (SC), and strict invariance model (ST) were considered to assess the consistency of the factorial structure across gender groups. Additionally, measurement invariance levels were evaluated following the recommendations of Cheung and Rensvold [66], where ΔCFI ≤ .01 and ΔRMSEA ≤ .01 indicate invariance.

Finally, to minimize Type I error when evaluating the UCLA-R items and gender differences, a univariate analysis of variance (ANOVA) was conducted for each item on the scale. Eta squared ($\eta^2$) was used to estimate effect sizes in the comparison between male and female groups, considering values for small effects (.01−.05) and medium effects (.06−.13), according to Levine and Hullett [67].

## Results

### Exploratory Factor Analysis (EFA)

**Data matrix adequacy.** The adequacy of the data matrix for the complete scale was assessed, with a KMO index of.870, which indicates excellent sampling adequacy for factor analysis. Additionally, Bartlett's test of sphericity was significant ($\chi^2$ = 3003.579, p < .001), confirming the suitability of the data for factor analysis.For factor extraction, in addition to the parallel analysis based on principal components, following Lasgaard [61], Cattell's scree plot [68] and Horn's parallel analysis [69,70] were considered, both of which suggested the retention of two factors. The latter analysis determined eigenvalues of 6.463 for factor 1 and 2.428 for factor 2, both of which were higher than the simulated data (1.412 and 1.339, respectively). Together, the two factors explained 39.1% of the total variance.

In Table 2, mean scores are observed to range between 2.045 (U7) and 3.073 (U5), with typical deviations between 0.801 and 1.028, which indicates moderate variability in responses. The normality of each item's distribution was evaluated through asymmetry and kurtosis coefficients. Asymmetry values ranged between −0.495 and 0.462, while kurtosis values were situated between −1.142 and −0.361. According to Mardia et al.'s criteria [58], the data showed an approximately normal distribution. Additionally, the oblique rotation (Oblimin) is presented, where factor loadings indicated that items U19, U10, U1, U20, U16, U5, U6, and U15 strongly loaded onto Factor 1. On the other hand, Factor 2 showed higher loadings for items U14, U11, U2, U18, and U8, while U7, U12, U3, U4, U13, and U17 were associated with lower factor loadings. Finally, item U9, which states "I am an extroverted person," showed a factor loading below the established threshold of 0.30 and was therefore removed [71]. This result may be attributed to the low conceptual alignment of the item with the construct of loneliness, as it assesses a personality trait (extroversion) rather than a direct perception of social isolation or emotional disconnection. Previous studies have similarly found that items not directly reflecting loneliness tend to exhibit weak or inconsistent factor loadings in adolescent populations [49,72].

After the removal of item U9, which did not load on any factor according to the established criterion (≥ .30) from the previous analysis, Table 3 shows a total explained variance of 40.8%, where Factor 1 (8 items) accounts for 23.7% and Factor 2 (11 items) accounts for 17.1%. These results indicate a more stable and appropriate factorial structure for measuring the construct of loneliness.

### Confirmatory Factor Analysis (CFA) using the nB sample

To determine the factorial structure of the UCLA-R in adolescents, goodness-of-fit indices were compared across three models: the unidimensional model (M1) with 20 items from the full scale proposed by Lasgaard [61], the bifactorial model (M2) suggested by the EFA (19 items), and the three-factor model (M3) with 20 items proposed by Kwiatkowska et al. [72], all applied to adolescent populations.

The results are presented in Table 4, where M2 demonstrates a significant advantage in terms of overall model fit compared to M1 and M3. M2 yielded a CMIN/DF of 1.89 (≤ 3), a CFI of.936, exceeding the acceptable threshold (≥.90),

**Table 2. Factor Model Pattern Matrix and Measures of Central Tendency, Dispersion, Skewness, and Kurtosis of the 20-Item UCLA-R for Adolescents.**

| Items | Mean | SD | Skew | Kurt | F1 | F2 |
|---|---|---|---|---|---|---|
| U19 | 2.930 | .911 | −.285 | −.978 | .816 | |
| U10 | 3.017 | .849 | −.363 | −.798 | .797 | |
| U1 | 2.911 | .821 | −.199 | −.756 | .787 | |
| U20 | 2.819 | .968 | −.151 | −1.142 | .751 | |
| U16 | 2.772 | .914 | −.017 | −1.062 | .714 | |
| U5 | 2.942 | .997 | −.495 | −.894 | .712 | |
| U6 | 2.660 | .888 | −.096 | −.750 | .662 | |
| U15 | 2.510 | .912 | .104 | −.803 | .647 | |
| U14 | 2.181 | .929 | .367 | −.727 | | .746 |
| U11 | 2.217 | .956 | .383 | −.769 | | .712 |
| U2 | 2.387 | .993 | .109 | −1.033 | | .634 |
| U18 | 2.557 | .892 | .065 | −.764 | | .631 |
| U8 | 2.315 | .801 | .225 | −.361 | | .494 |
| U7 | 2.045 | .851 | .462 | −.432 | | .471 |
| U12 | 2.270 | .870 | .395 | −.441 | | .463 |
| U3 | 2.195 | .904 | .405 | −.572 | | .450 |
| U4 | 2.368 | .988 | .153 | −1.003 | | .436 |
| U13 | 2.705 | .987 | −.046 | −1.133 | | .337 |
| U17 | 2.248 | .959 | .386 | −.763 | | .321 |
| U9 | 2.507 | .951 | .009 | −.916 | | |
| Mardia's Multivariate Test | | | 2.191.232 | 12.634 | | |
| Mardia's p-value | | | < .001 | < .001 | | |
| Explained Variance Percentage | | | | | 22.9% | 16.2%. |
| Total Variance Explained | | | | | 39.1%. | |

Note. SD, standard deviation; Skew, Skewness; Kurt, Kurtosis; F1, factor 1; F2, factor 2.

and a TLI of.927, also within the appropriate range (≥.90). Regarding SRMR (.048) and RMSEA (.050), both values fall within the recommended ranges, supporting the suitability of this model. In comparison, M1 exhibited a poorer fit across the CMIN/DF, CFI, and TLI indices, all of which were below acceptable thresholds. Meanwhile, although M3 displayed adequate values for CMIN/DF, CFI, and TLI, it did not surpass M2 in terms of overall model fit.

Regarding the results, M2 exhibited the lowest AIC (16348.614) and BIC (16577.730) values, indicating a more parsimonious model compared to the other two models (AIC for M1 = 17625.225, BIC = 17858.224; AIC for M3 = 17326.754, BIC = 17571.404). Additionally, the internal consistency coefficients for M2 showed better values, with ω = 0.83 and α = 0.876, suggesting high internal reliability of the 19-item UCLA-R scale, in contrast to M1 and M3.

Additionally, the Cronbach's alpha values for the individual factors in M2 were high (F1α = .879, F2α = .760), supporting adequate internal consistency within each dimension. Regarding the stratified Cronbach's alpha, M2 showed a higher value (αs = .818), indicating a more precise estimate of the reliability of the multidimensional structure compared to M1, which, being unidimensional, does not report a stratified alpha, and M3, which showed some factor-specific alphas but presented notably low reliability for factor 3 (F3α = .427) and a lower stratified alpha (αs = .783) compared to M2, suggesting less consistency in measurement across the dimensions.

On the other hand, Fig 1 visually represents the bifactorial model (M2), which organizes the UCLA-R (19 items) into two latent factors, both influenced by a common variable, UCL. Regarding the factor loadings, they indicate a strong

**Table 3. Pattern Matrix of the 19-Item UCLA-R for Adolescents.**

| Items | F1 | F2 |
|---|---|---|
| U19 | .818 | |
| U10 | .801 | |
| U1 | .784 | |
| U20 | .752 | |
| U16 | .719 | |
| U5 | .711 | |
| U6 | .660 | |
| U15 | .636 | |
| U14 | | .747 |
| U11 | | .713 |
| U2 | | .634 |
| U18 | | .631 |
| U8 | | .492 |
| U7 | | .470 |
| U12 | | .462 |
| U3 | | .449 |
| U4 | | .436 |
| U13 | | .332 |
| U17 | | .327 |
| Percentage of variance | 23.7%. | 17.1%. |
| Total variance | 40.8% | |

*Note.* F1, factor 1; F2, factor 2.

**Table 4. Goodness-of-Fit Indices and Reliability of the Confirmatory Factor Analysis of the 19-Item UCLA-R for Adolescents.**

| Index | M1 | M2 | M3 |
|---|---|---|---|
| CMIN/DF | 3.82 | 1.89 | 2.06 |
| CFI | .777 | .936 | .918 |
| TLI | .751 | .927 | .906 |
| SRMR | .077 | .048 | .058 |
| RMSEA | .089 | .050 | .054 |
| AIC | 17625.225 | 16348.614 | 17326.754 |
| BIC | 17858.224 | 16577.730 | 17571.404 |
| ωT | .513 | .83 | .781 |
| αT | .538 | .876 | .874 |
| F1α | – | .879 | .746 |
| F2α | – | .760 | .852 |
| F3α | – | – | .427 |
| $α_s$ | – | .818 | .783 |

Note. M1, Model 1; M2, Model 2; M3, Model 3; CMIN/DF, chi-square ratio ($χ^2$) to degrees of freedom; CFI, Bentler's Comparative Fit Index; TLI, Tucker-Lewis Index; SRMR, Standardized Root Mean Square Residual; RMSEA, Root Mean Square Error of Approximation; AIC, Akaike Information Criterion; BIC, Bayesian Information Criterion; ωT, Overall omega coefficient; αT, Overall Cronbach's alpha; F1α, Cronbach's alpha for Factor 1; F2α, Cronbach's alpha for Factor 2; F3α, Cronbach's alpha for Factor 3; $α_s$, Stratified Cronbach's alpha.

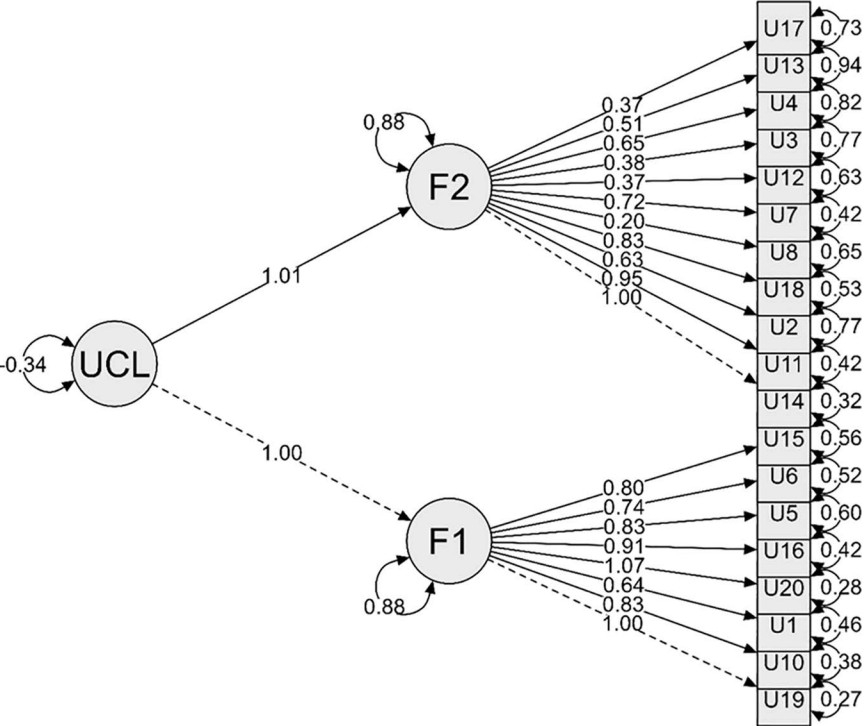

**Fig 1. Bifactorial structure of the 19-item UCLA-R scale for adolescents.**

relationship between the latent factors and the observed items. In this sense, F1 and F2 share a strong covariance of 0.88, indicating that, although they represent distinct dimensions of the loneliness construct, they have a significant proportion of shared explained variance. Additionally, the influence of UCL on both factors highlights the bifactorial structure, which aligns with the goodness-of-fit results, as they fall within the recommended ranges.

## Factorial invariance analysis

Regarding the factorial invariance between males and females in the total sample (N = 718), Table 5 shows a good model fit for both groups. Although the chi-square statistic is reported, it is important to consider that it is highly sensitive

**Table 5. Factorial Invariance by Gender of the 19-Item UCLA-R Scale for Adolescents.**

| Model | χ² | df | C-M | Δχ² | Δdf | CFI | ΔCFI | RMR | RMSEA | ΔRMSEA |
|---|---|---|---|---|---|---|---|---|---|---|
| Total sample | 448.442 | 150 | | | | .927 | | .042 | .053 | |
| Males | 297.749 | 150 | – | – | – | .913 | – | .054 | .058 | – |
| Females | 327.913 | 300 | – | – | – | .926 | – | .046 | .053 | – |
| MC | 625.662 | 300 | – | – | – | .921 | – | .049 | .055 | – |
| MM | 668.717 | 318 | MM-MC | 43.055 | 18 | .914 | −.007 | .058 | .055 | 0 |
| SC | 708.439 | 334 | SC-MM | 39.722 | 16 | .909 | −.005 | .060 | .056 | .001 |
| ST | 750.842 | 353 | ST-SC | 42.403 | 19 | .903 | −.006 | .060 | .056 | 0 |

Note: χ², chi-square test; df, degrees of freedom; C-M, factorial invariance model comparison; CFI, Bentler's Comparative Fit Index; RMR, Standardized Root Mean Square Residual; RMSEA, Root Mean Square Error of Approximation; Δ, change; Models: MC, configural model; MM, metric model; SC, scalar model; ST, strict model.

to sample size, often resulting in statistically significant outcomes even for models with good fit. Therefore, greater emphasis is placed on the comparative fit indices (CFI, TLI) and their differences (ΔCFI), as well as on the RMSEA, to evaluate model fit and factorial invariance. The observed values were $\chi^2 = 448.442$, df = 150, CFI = .927, RMR = .042, and RMSEA = .053. With respect to the gender analysis, both males and females demonstrated an acceptable model fit. For male adolescents, the values were $\chi^2 = 297.749$, df = 150, CFI = .913, RMR = .054, and RMSEA = .058; whereas for females, the values were $\chi^2 = 327.913$, df = 300, CFI = .926, RMR = .046, and RMSEA = .053. These results suggest that the model structure is consistent across both groups.

With regard to factorial invariance, the configural model (MC) demonstrated that the factorial structure is similar for both males and females, with $\chi^2 = 625.662$, df = 300, CFI = .921, RMR = .049, and RMSEA = .055. The metric model (MM) presented a difference of $\Delta\chi^2 = 43.055$ with Δdf = 18. Although this difference is significant, the values of ΔCFI = −.007 and ΔRMSEA = 0 fall within acceptable ranges, confirming that factor loadings are similar between genders. Meanwhile, the scalar model (SC) showed a difference of $\Delta\chi^2 = 39.722$, with Δdf = 16, ΔCFI = −.005, and ΔRMSEA = .001, indicating that the items are comparable between males and females. Finally, the strict model (ST) supports that measurement errors are similar between groups, with a difference of $\Delta\chi^2 = 42.403$, Δdf = 19, ΔCFI = −.006, and ΔRMSEA = 0, allowing for comparisons between males and females.

## Univariate Analysis of Variance (ANOVA)

In Table 6, the gender differences in the 19-item UCLA-R scale are presented. In general terms, females reported statistically significant differences (p < .05) in items U2, U4, U7, U11, U14, U15, and U18 compared to males. However, the effect size of these differences was small to moderate. The largest differences were found in items U4 (p = .001, η² = .02574448),

**Table 6. Gender Differences in the 19-Item UCLA-R Scale for Adolescents.**

| ITEMS | Male n = 288 (M ± SD) | Female n = 288 (M ± SD) | F | p | η² |
|---|---|---|---|---|---|
| U1 | 3.024 ± .832 | 2.926 ± .819 | 2.474 | .116 | .00344376 |
| U2 | 2.271 ± 1.027 | 2.498 ± .960 | 9.099 | .003 | .01254864 |
| U3 | 2.205 ± .953 | 2.174 ± .888 | .191 | .662 | .00026712 |
| U4 | 2.122 ± 1.041 | 2.451 ± .964 | 18.921 | .001 | .02574448 |
| U5 | 3.073 ± .973 | 2.974 ± 1.007 | 1.697 | .193 | .00073581 |
| U6 | 2.694 ± .909 | 2.640 ± .889 | .644 | .422 | .00087561 |
| U7 | 1.917 ± .826 | 2.084 ± .851 | 6.798 | .009 | .00940408 |
| U8 | 2.233 ± .817 | 2.335 ± .805 | 2.752 | .098 | .00382808 |
| U10 | 3.080 ± .866 | 3.056 ± .854 | .135 | .713 | .00018942 |
| U11 | 2.031 ± .877 | 2.326 ± .985 | 16,792 | .001 | .0229144 |
| U12 | 2.337 ± .915 | 2.214 ± .808 | 3.580 | .059 | .00497469 |
| U13 | 2.545 ± 1.028 | 2.679 ± 1.003 | 3.013 | .083 | .00419117 |
| U14 | 2.014 ± .887 | 2.321 ± .933 | 19.415 | .001 | .0263997 |
| U15 | 2.722 ± .948 | 2.463 ± .910 | 13.566 | .001 | .01859549 |
| U16 | 2.823 ± .922 | 2.777 ± .924 | .431 | .512 | .00060239 |
| U17 | 2.115 ± .939 | 2.219 ± .923 | 2.16 | .142 | .00300708 |
| U18 | 2.399 ± .878 | 2.565 ± .948 | 5.596 | .018 | .00775471 |
| U19 | 3.017 ± .928 | 2.958 ± .892 | .736 | .391 | .00102757 |
| U20 | 2.899 ± .952 | 2.849 ± .965 | .477 | .49 | .00066546 |

Note: n, absolute frequency of the sample; M, mean; SD, standard deviation; F, Fisher's statistic in ANOVA; p, significance level; η², eta squared.

U14 (p = .001, η² = .0263997), and U15 (p = .001, η² = .01859549), with a moderate effect size. Although these differences are statistically significant, the η² values indicate that the influence of gender on loneliness perception is relatively low and may be determined by various additional factors.

On the other hand, no significant differences between males and females (p < .05) were found in items U1, U3, U5, U6, U8, U10, U12, U13, U16, U17, U19, and U20, indicating that loneliness perception remains similar across both groups. These results are consistent with the EFA, CFA, and factorial invariance analyses, supporting the stability of the bifactorial structure in the sample of Ecuadorian adolescents.

## Discussion

The perception of loneliness among adolescents has significantly increased in recent decades, negatively affecting their overall well-being [73]. This situation has become a global issue [74] and has attracted the attention of both the scientific community and governmental organizations, considering it a public health concern [75]. In this context, and given the importance of studying loneliness during this stage of development, the present study focused on analyzing the psychometric properties of the UCLA-R Loneliness Scale in Ecuadorian adolescents through exploratory and confirmatory factor analysis, assess its internal consistency, and examine factorial invariance across genders, as its psychometric properties have not yet been clearly established in various contexts [32], and even less so in Ecuadorian adolescents.

In this study, the internal consistency and reliability of the scale were high, with values of α = .876 and ω = .83. These results are similar to those obtained in other validation studies conducted in different contexts. For instance, Ausín et al. [41] reported an α = .85 in a Spanish sample of older adults, while in Denmark, Lasgaard [61] obtained an α = .92 in adolescents. In Brazil, Fonsêca et al. [50] found an α = .93 in university students, and in Peru, Vera et al. [30] reported an α = .71 in Peruvian adolescents. These values indicate that the scale demonstrates good internal consistency both in adolescents and in other populations. Although few recent studies report omega coefficient values, some exceptions include Vera et al. [30], Lin et al. [43], and Ventura-León & Caycho [76], who obtained ω ≥ .70, a result consistent with the findings of this study.

The EFA and CFA analyses revealed the existence of two factors in the UCLA-R applied to Ecuadorian adolescents, which correspond to the dimensions of emotional loneliness and social loneliness. These findings are consistent with those of Lasgaard [61], who also identified two factors in the Danish version of the scale. These results align with previous research that has also identified multiple dimensions (two or three factors) in the scale. For instance, studies conducted in the general population and older adults [44,77] found multifactorial structures in the scale. Similarly, in both the Netherlands and Northern Ireland, Goossens et al. [49] and Shevlin et al. [78] reported a three-factor structure of the UCLA-R applied to adolescents. However, some researchers have reported a unidimensional structure in shorter versions of the scale. Among them, Vera et al. [30] and Lin et al. [43] found a single-factor solution, although not always in adolescent populations. This suggests that the factorial structure of the scale may vary depending on the population and cultural context, emphasizing the importance of continued research on its factorial structure in different contexts [43,79,80].

On the other hand, following the EFA, item U9 "I am an extroverted person" was excluded due to its low factor loading [71]. This decision is supported by Bethell et al. [81], who argue that extraversion is a relatively stable personality trait that does not necessarily reflect specific states of loneliness or social disconnection. Furthermore, studies by Buecker et al. [82] and Matthews et al. [83] conclude that, although there is a negative relationship between extraversion and loneliness, it is of moderate magnitude and does not capture concrete affective-social experiences. In this regard, previous validations of the UCLA scale in adolescents have also reported difficulties with items related to general personality traits [49,72].

Regarding the CFA values, the bifactorial model (19 items) explains 40.8% of the variance in the scale. This finding is similar to adaptations of shorter versions of the scale reported in other studies, such as the 10-item version by Russell [45] and the 6-item version by Neto [84], both designed to enhance measurement efficiency without compromising reliability.

Additionally, even more reduced versions, such as the three-item version by Hughes et al. [47], reinforce the idea of using shorter versions of the questionnaire in contexts where a rapid yet accurate assessment is required, without losing validity or reliability [43,45].

Regarding the goodness-of-fit indices obtained in this study, the values were satisfactory: CFI = .936, TLI = .927, RMSEA = .050, and SRMR = .048. These results are comparable to studies that employed both full and short versions of the scale, such as those by Lin et al. [43], Vera et al. [30], Wongpakaran et al. [42], and Ozdemir & Tan [79]. This supports the idea that both full and abbreviated versions can exhibit a good fit depending on the context and the evaluated population.

In this regard, the present study supports previous research that conceptualizes loneliness as a multidimensional construct [44,49,61]. This is consistent with studies conducted in diverse populations, such as adolescents in Denmark [61], older adults [44], and young people in Northern Ireland [78], indicating that the bifactorial nature of loneliness remains consistent across different cultures and age groups. Therefore, the findings of this study provide new evidence from a Latin American context, specifically among Ecuadorian adolescents a population that has been scarcely considered in the psychometric validation literature [30].

Regarding the internal consistency and reliability of the scale across gender groups the results were adequate. Although reliability was slightly higher in females, this result suggests that adolescents experience and express loneliness differently depending on their sex. This finding is consistent with previous research indicating differences in the experience of loneliness, where females tend to report higher levels of loneliness compared to males, although males are also significantly affected [28]. This reinforces the importance of continuing to study the UCLA-R across different age groups and populations.

On the other hand, the validity results of the 19-item UCLA-R scale in this study support its use for both males and females, as no significant differences were found in its factorial structure by gender in adolescents. This finding aligns with the results of Neto [84], Lin et al. [43], Vera et al. [30], McDonald et al. [85], and Bajaj & Kaur [86], who also did not report sex differences in the structure of the scale.

From a practical standpoint, the validated 19-item version of the UCLA-R Loneliness Scale provides education and mental health professionals with a reliable and culturally sensitive tool for assessing loneliness in adolescents. Early detection of loneliness is crucial, as it has been linked to a wide range of physical and psychological health problems, including depression, anxiety, and even increased morbidity and mortality [5,11,13]. In educational settings, the scale can support the development of interventions aimed at strengthening peer relationships and promoting social integration [27,28]. In the field of public health, it offers empirical support for strategies addressing loneliness as a growing concern during adolescence [74,75].

## Limitations

Despite the relevance of the findings, some limitations of this study have been identified. The sample consisted of school-attending adolescents from a specific region of Ecuador, which may limit the generalizability of the results to other population contexts. Moreover, the use of online surveys may have introduced response biases, particularly among adolescents with limited digital literacy or restricted internet access. Therefore, it is suggested that future studies administer the scale in printed format. Another limitation is the sample size, which, although statistically adequate, may not fully represent the diverse demographic and sociocultural subgroups in Ecuador. Future research should consider increasing sample diversity and conducting longitudinal studies to assess the scale's stability and evaluate changes in loneliness over time.

Although this study did not conduct analyses of convergent or divergent validity, this was due to the lack of standardized instruments for related constructs validated in Ecuadorian adolescents at the time of data collection. It is recommended that future studies include such analyses by incorporating measures of related psychological constructs (e.g., depression, social support) to further corroborate the construct validity of the scale.

## Conclusion

It is concluded that the 19-item version of the UCLA-R Loneliness Scale is a valid, reliable, and culturally appropriate tool for assessing perceived loneliness in Ecuadorian adolescents, with no gender differences, supporting its use among both male and female individuals within this age group. These findings reinforce its relevance as a suitable instrument for the study of loneliness in various adolescent contexts, including psychological, educational, and clinical settings. Furthermore, its validation within the Ecuadorian context highlights the importance of adapting and verifying psychological instruments in accordance with the sociocultural particularities of each population.

## Acknowledgments

The authors would like to thank to Universidad Técnica Particular de Loja – Ecuador. For its institutional support and for covering the processing costs of this article.

## Author contributions

**Conceptualization:** Katy Maricela Chamba Leiva.

**Data curation:** Wilson Guillermo Siguenza Campoverde.

**Formal analysis:** Katy Maricela Chamba Leiva.

**Methodology:** Wilson Guillermo Siguenza Campoverde.

**Supervision:** Katy Maricela Chamba Leiva.

**Validation:** Wilson Guillermo Siguenza Campoverde.

**Writing – original draft:** Katy Maricela Chamba Leiva.

**Writing – review & editing:** Wilson Guillermo Siguenza Campoverde.

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
