## [Decision Letter · Decision Letter 0]

14 May 2025

Dear Dr. Siguenza,

Thank you for submitting your manuscript to PLOS ONE. After careful consideration, we feel that it has merit but does not fully meet PLOS ONE’s publication criteria as it currently stands. Therefore, we invite you to submit a revised version of the manuscript that addresses the points raised during the review process.

**The two reviewers have raised useful comments on your manuscript. Please consider their comments thoroughly and work on a revised manuscript based on their comments. **

We look forward to receiving your revised manuscript.

Kind regards,

Ted C.T. Fong, Ph.D.

Academic Editor

PLOS ONE

**Journal Requirements:**

1. When submitting your revision, we need you to address these additional requirements. Please ensure that your manuscript meets PLOS ONE's style requirements, including those for file naming. The PLOS ONE style templates can be found at https://journals.plos.org/plosone/s/file?id=wjVg/PLOSOne_formatting_sample_main_body.pdf and https://journals.plos.org/plosone/s/file?id=ba62/PLOSOne_formatting_sample_title_authors_affiliations.pdf 2. Please ensure that the title is match in the manuscript and the cover letter. 3. We note that the grant information you provided in the ‘Funding Information’ and ‘Financial Disclosure’ sections do not match.  When you resubmit, please ensure that you provide the correct grant numbers for the awards you received for your study in the ‘Funding Information’ section. 4. Please note that your Data Availability Statement is currently missing the repository name. If your manuscript is accepted for publication, you will be asked to provide these details on a very short timeline. We therefore suggest that you provide this information now, though we will not hold up the peer review process if you are unable. 5. When completing the data availability statement of the submission form, you indicated that you will make your data available on acceptance. We strongly recommend all authors decide on a data sharing plan before acceptance, as the process can be lengthy and hold up publication timelines. Please note that, though access restrictions are acceptable now, your entire data will need to be made freely accessible if your manuscript is accepted for publication. This policy applies to all data except where public deposition would breach compliance with the protocol approved by your research ethics board. If you are unable to adhere to our open data policy, please kindly revise your statement to explain your reasoning and we will seek the editor's input on an exemption. Please be assured that, once you have provided your new statement, the assessment of your exemption will not hold up the peer review process. 6. We note that there is identifying data in the Supporting Information file “Completa Ucla”. Due to the inclusion of these potentially identifying data, we have removed this file from your file inventory. Prior to sharing human research participant data, authors should consult with an ethics committee to ensure data are shared in accordance with participant consent and all applicable local laws. Data sharing should never compromise participant privacy. It is therefore not appropriate to publicly share personally identifiable data on human research participants. The following are examples of data that should not be shared: -Name, initials, physical address-Ages more specific than whole numbers-Internet protocol (IP) address-Specific dates (birth dates, death dates, examination dates, etc.)-Contact information such as phone number or email address-Location data-ID numbers that seem specific (long numbers, include initials, titled “Hospital ID”) rather than random (small numbers in numerical order) Data that are not directly identifying may also be inappropriate to share, as in combination they can become identifying. For example, data collected from a small group of participants, vulnerable populations, or private groups should not be shared if they involve indirect identifiers (such as sex, ethnicity, location, etc.) that may risk the identification of study participants. Additional guidance on preparing raw data for publication can be found in our Data Policy (https://journals.plos.org/plosone/s/data-availability#loc-human-research-participant-data-and-other-sensitive-data) and in the following article: http://www.bmj.com/content/340/bmj.c181.long. Please remove or anonymize all personal information (<specific identifying information in file to be removed>), ensure that the data shared are in accordance with participant consent, and re-upload a fully anonymized data set. Please note that spreadsheet columns with personal information must be removed and not hidden as all hidden columns will appear in the published file. 7. Please include captions for your Supporting Information files at the end of your manuscript, and update any in-text citations to match accordingly. Please see our Supporting Information guidelines for more information: http://journals.plos.org/plosone/s/supporting-information.

Reviewers' comments:

Reviewer's Responses to Questions

**Comments to the Author**

1. Is the manuscript technically sound, and do the data support the conclusions?

Reviewer #1: Partly

Reviewer #2: Partly

2. Has the statistical analysis been performed appropriately and rigorously?

Reviewer #1: Yes

Reviewer #2: Yes

3. Have the authors made all data underlying the findings in their manuscript fully available?

Reviewer #1: Yes

Reviewer #2: Yes

4. Is the manuscript presented in an intelligible fashion and written in standard English?

Reviewer #1: Yes

Reviewer #2: Yes

**Reviewer #1:**  I had the pleasure of participating in the review process of a manuscript entitled "Factorial Structure, Validity, and Sex Differences of the UCLA-R 2 Loneliness Scale in Ecuadorian Adolescents."

The present manuscript, like any scientific paper, is subject to improvement, and I request the authors to make necessary corrections before acceptance for publication.

General comments:

The manuscript aimed to validate the UCLA-R 2 Loneliness Scale in Ecuadorian adolescents by assessing its factor structure, validity and gender invariance.

The introduction is well written and structured, providing the necessary literature review in relation to the study objective.

The methods are clear and relevant.

The discussion supports the results and conclusions.

Specific comments:

- Title: in my humble opinion, it should be better if you use “Gender Invariance” instead of “sex differences”. If you are ok, please change the expression throughout the manuscript.

- Abstract:

• Typos: “objective” instead of “objetive”.

• The objective should provide more information about the different analyses assessed to validate the scale.

• The method is too short and doesn't contain enough information, try enriching it with the necessary and appropriate information (e.g., average age and standard deviation, socio-demographic variables, sample proportions for conducting the EFA and CFA...).

• Results: provide findings from the EFA.

- Introduction:

• As I mentioned earlier, the introduction is clear and well structured. However, I recommend that the authors add a paragraph about the need for cross-cultural validation. Why is it important to use a validated scale in the local language?

- Materials and methods:

• Procedure:

- I must admit that the use of online surveys has several limitations (depending on the study). So, my question is: how did you try to limit respondent bias? Did you use a guide for using online surveys (e.g., CHERRIES (Eysenbach, 2012)? If not, try to explain your own procedure.

• Results:

- I suggest deleting table 2, since you've included the necessary information in the text, otherwise it'll be a repetition.

- Please provide findings related to the normality of the distribution assessed in table 3 by Skewness and Kurtosis and add it to table’s title.

• Discussion:

- I recommend adding paragraphs on practical and theoretical implications of the study.

• Limitations:

- Several other limitations should be added such as the bias on the use of online surveys, or the limited sample size.

- Explain why you didn’t conduct convergent or divergent validity.

• Conclusion:

- - I'm requesting that you rewrite the section, as you've just reported the results. The conclusion should be limited to an answer to the main objective of the study, so it's not necessary to report the GFI results.

**Reviewer #2: ** First, the method and results section of the scale adaptation study is very detailed and well presented. However, although the study was conducted in a large sample, it is an adaptation study and I am not sure that it meets the journal's requirement in this direction. My comments for the authors are as follows:

In line 20, I think the sentence means that the bifactor model does not change between genders in terms of measurement invariance. It could be written a little more clearly.

In line 102-103, Can you give some brief information about this methodology, is it like a simple random sampling method?

In line 216-217, Explanations can be given as to why this item does not load on any factor.

In table 5, Calculating alpha reliability for each dimension and calculating the stratified alpha coefficient for the entire scale may be more accurate in providing the lower limit of alpha reliability for multidimensional scales.

In line 274, You can point out with reference that it is more accurate to look at the difference between these indices, as the chi-squared statistic is affected by sample size.

In line 338, It was stated that one item was removed because it did not load sufficiently on any of the factors. It would have been good to explain why this item did not work with adolescents. Has a similar situation occurred with the same item in other adaptation studies?

**Do you want your identity to be public for this peer review?** For information about this choice, including consent withdrawal, please see our Privacy Policy

Reviewer #1: **Yes: ** Amayra Tannoubi

Reviewer #2: No

---

## [Author Response · Author response to Decision Letter 1]

27 May 2025

Response to the Editor’s Comments

1. Comment: Ensure your manuscript complies with PLOS ONE’s style requirements, including file naming conventions.

Response: The manuscript has been reviewed and adjusted according to PLOS ONE guidelines.

2. Comment: Ensure the title matches that of the manuscript and the cover letter.

Response: The title in the cover letter has been adjusted accordingly.

3. Comment: We noted that the grant information provided in the "Funding Information" and "Financial Disclosure" sections does not match.

Response: The issue regarding the financial grant has been clarified.

4. Comment: The repository name is missing in your Data Availability Statement.

Response: The name of the data repository has been added.

5. Comment: In the submission form, you indicated that the data would be made available upon acceptance.

Response: The anonymized database has been made openly accessible for general use.

6. Comment: We observed that the supplementary file "Complete Ucla" contains identifiable data.

Response: The database has been adjusted and anonymized.

7. Comment: Include the titles of your supplementary files at the end of your manuscript and update in-text citations to match.

Response: No supplementary files are included.

Response to Reviewer 1

1. Comment (Title): In my humble opinion, "Gender invariance" would be preferable to "sex differences." If agreeable, please revise the terminology throughout the manuscript.

Response: The suggestion has been considered, and the title and relevant sections of the manuscript have been updated as requested.

2. Comment (Abstract): Typographical error: "objective" instead of "objective."

Response: The typographical error has been corrected.

3. Comment (Abstract): The objective should provide more details about the different analyses conducted to validate the scale.

Response: The objective has been revised as suggested.

4. Comment (Methods): The methods section is too brief and lacks sufficient detail. Enrich it with necessary information (e.g., mean age and standard deviation, sociodemographic variables, sample proportions for EFA and CFA).

Response: The section has been expanded as requested.

5. Comment (Results): Provide the findings from the EFA.

Response: The EFA findings have been included.

6. Comment (Introduction): Add a paragraph on the need for cross-cultural validation. Why is it important to use a scale validated in the local language?

Response: A paragraph addressing this has been added.

7. Comment (Procedure): How did you attempt to limit respondent bias? Did you use a guideline for online surveys (e.g., CHERRIES, Eysenbach, 2012)? If not, explain your procedure.

Response: The procedure to limit bias in online surveys has been added.

8. Comment (Results): I suggest removing Table 2, as the necessary information is already included in the text. Otherwise, it would be redundant.

Response: Table 2 has been removed, and the numbering of subsequent tables has been adjusted.

9. Comment (Results): Provide findings related to the normality of the distribution (Skewness and Kurtosis) evaluated in Table 3 and add them to the table title.

Response: A paragraph addressing skewness, kurtosis, mean, and standard deviation has been added.

10. Comment (Discussion): I recommend adding paragraphs on the practical and theoretical implications of the study.

Response: Paragraphs on practical and theoretical implications have been included.

11. Comment (Limitations): Additional limitations should be added, such as bias in online surveys or the limited sample size. Explain why convergent or divergent validity was not assessed.

Response: Additional limitations and explanations have been provided.

12. Comment (Conclusion): Please rewrite this section, as it currently reiterates results. The conclusion should focus on answering the primary objective of the study, so reporting GFI results is unnecessary.

Response: The conclusion has been revised accordingly.

Response to Reviewer 2

1. Comment (Line 20): The sentence suggests that the bifactor model does not vary between genders in terms of measurement invariance. It could be phrased more clearly.

Response: The text has been adjusted for clarity.

2. Comment (Lines 102–103): Could you briefly explain this methodology? Is it similar to simple random sampling?

Response: The methodology has been clarified.

3. Comment (Lines 216–217): Provide explanations for why this item did not load onto any factor.

Response: An explanation has been added.

4. Comment (Table 5): Calculating alpha reliability for each dimension and stratified alpha coefficients for the entire scale would provide a more precise lower bound of alpha reliability for multidimensional scales.

Response: The requested values have been included.

5. Comment (Line 274): You could reference why it is more accurate to observe differences between these indices, as the chi-square statistic is affected by sample size.

Response: A reference and explanation have been added.

6. Comment (Line 338): You mentioned removing an item because it did not sufficiently apply to any factor. It would be helpful to explain why this item did not work with adolescents. Has this issue been observed in other adaptation studies?

Response: A paragraph addressing this has been added to the discussion.

---

## [Decision Letter · Decision Letter 1]

27 Jun 2025

Factorial structure, validity, and gender invariance of the UCLA-R Loneliness Scale in Ecuadorian adolescents

PONE-D-25-08122R1

Dear Dr. Siguenza,

We’re pleased to inform you that your manuscript has been judged scientifically suitable for publication and will be formally accepted for publication once it meets all outstanding technical requirements.

Kind regards,

Ted C.T. Fong, Ph.D.

Academic Editor

PLOS ONE

Additional Editor Comments (optional):

Reviewers' comments:

Reviewer's Responses to Questions

**Comments to the Author**

Reviewer #1: All comments have been addressed

2. Is the manuscript technically sound, and do the data support the conclusions?

Reviewer #1: Yes

3. Has the statistical analysis been performed appropriately and rigorously?

Reviewer #1: Yes

4. Have the authors made all data underlying the findings in their manuscript fully available?

Reviewer #1: Yes

5. Is the manuscript presented in an intelligible fashion and written in standard English?

Reviewer #1: Yes

Reviewer #1: After the first round of revision, I thank the authors for their rigorous work yo improve the quality of the manuscript. The paper is now suitable for publication. Congrats!!

**Do you want your identity to be public for this peer review?** For information about this choice, including consent withdrawal, please see our Privacy Policy

Reviewer #1: **Yes: ** Amayra Tannoubi

---

## [Editor Report · Acceptance letter]

PONE-D-25-08122R1

PLOS ONE

Dear Dr. Siguenza Campoverde,

I'm pleased to inform you that your manuscript has been deemed suitable for publication in PLOS ONE. Congratulations! Your manuscript is now being handed over to our production team.

Kind regards,

on behalf of

Dr. Ted C.T. Fong

Academic Editor

PLOS ONE